# Structure Analysis and Anti-Tumor and Anti-Angiogenic Activities of Sulfated Galactofucan Extracted from *Sargassum thunbergii*

**DOI:** 10.3390/md17010052

**Published:** 2019-01-11

**Authors:** Weihua Jin, Wanli Wu, Hong Tang, Bin Wei, Hong Wang, Jiadong Sun, Wenjing Zhang, Weihong Zhong

**Affiliations:** 1College of Biotechnology and Bioengineering, Zhejiang University of Technology, Hangzhou 310014, China; jinweihua@zjut.edu.cn (W.J.); 15726816712@163.com (W.W.); 17816035715@163.com (H.T.); 2Collaborative Innovation Center of Yangtze River Delta Region Green Pharmaceuticals & College of Pharmaceutical Sciences, Zhejiang University of Technology, Hangzhou 310014, China; binwei@zjut.edu.cn (B.W.); hongw@zjut.edu.cn (H.W.); 3Department of Biomedical and Pharmaceutical Sciences, College of Pharmacy, University of Rhode Island, Kingston, RI 02881, USA; jiadong.sun@nih.gov; 4Laboratory of Bioorganic Chemistry, National Institute of Diabetes and Digestive and Kidney Diseases (NIDDK), National Institutes of Health, Bethesda, MD 20878, USA; 5College of Life and Environmental Science, Wenzhou University, Wenzhou 325035, China

**Keywords:** sulfated galactofucan, fucoidan, anti-angiogenic activity, anti-tumor activity, *Sargassum thunbergii*

## Abstract

Sulfated galactofucan (ST-2) was obtained from *Sargassum thunbergii*. It was then desulfated to obtain ST-2-DS, and autohydrolyzed and precipitated by ethanol to obtain the supernatant (ST-2-S) and precipitate (ST-2-C). ST-2-C was further fractionated by gel chromatography into two fractions, ST-2-H (high molecular weight) and ST-2-L (low molecular weight). Mass spectrometry (MS) of ST-2-DS was performed to elucidate the backbone of ST-2. It was shown that ST-2-DS contained a backbone of alternating galactopyranose residues (Gal)_n_ (n ≤ 3) and fucopyranose residues (Fuc)_n_. In addition, ST-2-S was also determined by MS to elucidate the branches of ST-2. It was suggested that sulfated fuco-oligomers might be the branches of ST-2. Compared to the NMR spectra of ST-2-H, the spectra of ST-2-L was more recognizable. It was shown that ST-2-L contain a backbone of (Gal)_n_ and (Fuc)_n_, sulfated mainly at C4 of Fuc, and interspersed with galactose (the linkages were likely to be 1→2 and 1→6). Therefore, ST-2 might contain a backbone of (Gal)_n_ (n ≤ 3) and (Fuc)_n_. The sulfation pattern was mainly at C4 of fucopyranose and partially at C4 of galactopyranose, and the branches were mainly sulfated fuco-oligomers. Finally, the anti-tumor and anti-angiogenic activities of ST-2 and its derivates were determined. It was shown that the low molecular-weight sulfated galactofucan, with higher fucose content, had better anti-angiogenic and anti-tumor activities.

## 1. Introduction

Fucoidans are one type of water-soluble polysaccharide and are synthesized by brown algae. They are heteropolysaccharides containing not only fucose but also residues of galactose, mannose, rhamnose, glucuronica acid, glucose, xylose, and others. Based on the monosaccharides, fucoidans are divided into two types: sulfated heteropolysaccharides and sulfated galactofucans/sulfated fucans, which are the most well-known of the fucoidans.

There are many structures of sulfated galactofucans/sulfated fucans. To the best of our knowledge they can be divided into four types according to their backbones: (1) with the backbone of a 3-linked α-l-Fuc, the sulfation pattern could be at C2 and/or C4 [1,2,3,4], while (Fuc)_n_, sulfated (Fuc)_n_, (Gal)_n_, and/or sulfated (Gal)_n_ could be branches [5]; (2) with the backbone of alternating 3-linked α-L-Fuc and 4-linked α-L-Fuc, sulfation could also be at C2, C3, and/or C4 [6,7]; (3) with the backbone of 3-linked α-l-Fuc, sulfation could be at C4 and interspersed with α-D-Gal [8]; (4) With the backbone of 6-linked β-D-Galp and/or 2-linked Manp, branches of terminal galactose and fucose are attached at C4 or C2 [9].

Malignant tumors are one of the diseases that pose a severe threat to a person’s life. Unfortunately, the drugs can produce many side effects. Thus, it is important to discover and develop new, effective, nontoxic compounds from natural sources. Fucoidan, belonging to a family of sulfated heteropolysaccharides, has numerous biological activities. There are many good reviews [10,11,12,13,14,15,16,17,18,19,20] of these biological activities. In previous studies, fucoidans from *Sargassum thunbergii* [21,22,23,24] as well as other brown algae [25,26,27] have shown potent anti-tumor activities.

In this study, we attempt to elucidate the structural features of sulfated galactofucan from *Sargassum thunbergii* by desulfation and autohydrolysis. In addition, the anti-tumor and anti-angiogenic activities of sulfated galactofucan and its derivatives are determined.

## 2. Results 

### 2.1. Chemical Compositions and Structrual Analyses of Sulfated Galactofucan and Its Derivates

Polysaccharide, ST-2, was obtained by anion exchange chromatography on a DEAE-Bio Gel Agarose FF gel (6 cm × 40 cm) with elution by 2 M NaCl. The chemical composition of ST-2 showed that it contained 31.36% fucose (Fuc), 23.01% sulfate, 75.48% total sugar, and 2.43% uronic acid (UA). The molar ratio of monosaccharides was 0.04:0.03:0.05:0.04: 0.41:0.02:1.00 (Man:Rha:GlcA:Glc:Gal:Xyl:Fuc). The molecular weight was 143.0 and 36.7 kDa. Thus, it was concluded that ST-2 was mainly sulfated galactofucan. After desulfation, ST-2-DS contained 48.69% Fuc, 9.00% sulfate, 95.78% total sugar, and 3.65% UA. The molar ratio of monosaccharides was 0.08:0.06:0.07:0.02:0.30:0.03:1.00 (Man:Rha:GlcA:Glc:Gal:Xyl:Fuc). The molecular weight was 176.0, 26.0 and 2.3 kDa. It was apparent that the molar ratio of Gal to Fuc decreased by approximately 0.1 (from 0.4 to 0.3), while the molar ratios of Man and GlcA to Fuc increased after desulfation. It was shown that preferential destruction of fucan moieties during the desulfation process was observed for sulfated galactofucans from *Saccharina latissima* [1] and *Kjellmaniella crsaaifolia* [28]. However, it was also reported that the ratio of Gal to Fuc was unaffected for sulfated galactofucans from *Sargassum polycystum* [8] and *Sargassum fusiforme* [29]. Thus, it was proposed that this phenomenon was attributed to the location of Gal residues. Gal residues were located on the backbone leading to a constant ratio (Gal to Fuc), while Gal residues were located at the branches, leading to the change in the ratio (Gal to Gal). 

The IR spectra of ST-2 (Figure 1A) and ST-2-DS (Figure 1B) are provided in Figure 1. The main difference between ST-2 and ST-2-DS was the presence of bands detected at 1252 and 842 cm^−1^, which represented the characteristic bands of sulfate esters. The very intense and broad band at 1252 cm^−1^ was attributed to the asymmetric O=S=O stretching vibration of sulfate esters, with some contribution from COH, CC, and CO vibrations; while the band at 842 cm^−1^ was assigned to sulfate groups at the axial C4 positions, suggesting that sulfation occurred at C4. The absence of the band at 820 cm^−1^ was attributed to the C-O-S bending vibration of the sulfate substituents at the equatorial C2 or C3 positions, indicating that few or no sulfations occurred at C2 or C3 [30,31]. Thus, it was concluded that the sulfations of ST-2 occurred mainly at the C4 position. 

Mass spectroscopy (MS) was performed to determine the chemical composition. MS is widely used for the analysis of heteropolysaccharides due to its speed, sensitivity, and accuracy. However, polysaccharides with high molecular weights or a high degree of sulfate substitutions are not suitable. Thus, ST-2 was partly desulfated. Figure 2 shows the negative-ion mode electrospray ionization mass spectrometry (ESI-MS) spectrum of ST-2-DS. ST-2-DS contained a series of mono-sulfated fuco-oligosaccharides, methyl glycosides of mono-sulfated fuco-oligosaccharides. Moreover, the mixture also contained methyl glycosides of mono-sulfated and di-sulfated galacto-fuco-oligosaccharides (hexose was found to be galactose-based on the monosaccharide analysis). The proposed compositions of the ions are summarized in Table 1. 

To elucidate the structural features of the proposed compositions, we performed mass spectrometry in tandem with collision-induced dissociation tandem mass spectrometry (ESI-CID-MS/MS). Figure 3A shows the fragmentation pattern for the ion at *m*/*z* 403.089 (−1), which corresponded to [Me Fuc_2_(SO_3_H)-H]^−^. The identification of B-type ions at *m*/*z* 225.005 (−1) and 371.059 (−1) indicated that the sulfate was located at the non-reducing end, while the Y-type ion at *m*/*z* 257.030 (−1) suggested that the sulfate was located at the reducing end. Therefore, we concluded that Me (Fuc)_2_(SO_3_H) represented a mixture of the isomers Fuc(SO_3_H) → Fuc-OMe and Fuc → Fuc(SO_3_H)-OMe. 

In addition to the sulfated fuco-oligosaccharides and methyl glycosides of sulfated fuco-oligosaccharides, we also detected methyl glycosides of sulfated galacto-fuco-oligosaccharides. Therefore, ESI-CID-MS/MS was performed to elucidate the structural features of the sulfated galacto-fuco-oligosaccharides. The ESI-CID-MS/MS spectrum of the ion at *m*/*z* 565.141 (−1) ([Me Gal Fuc_2_ (SO_3_H)-H]^−^) is shown in Figure 3B. The ions at *m*/*z* 225.004 (−1), 240.998 (−1), and 257.029 (−1) corresponded to [Fuc(SO_3_H)-H]^−^, [Gal(SO_3_H)-H]^−^, and [Me Fuc(SO_3_H)-H]^−^, respectively. This result indicated that the sulfate was located at the non-reducing and reducing ends. Additionally, the ions at *m*/*z* 371.058 (−1), 387.052 (−1), 403.083 (−1), and 419.078 (−1) derived from the loss of the methyl glycoside of galactose (−194 Da), methyl glycoside of fucose (−178 Da), galactopyranose (−162 Da), and fucopyranose (−146 Da), indicating that the reducing end contained Gal-OMe and Fuc-OMe residues, while the non-reducing end contained Fuc and Gal. Moreover, the ion at *m*/*z* 533.106 represented the loss of methanol (32 Da). Based on this result (i.e., the sulfate was located at both the reducing and non-reducing ends), we propose that the ion at *m*/*z* 565.141 (−1) ([Me Gal Fuc_2_ (SO_3_H)-H]^−^) was also a mixture of isomers: Fuc(SO_3_H) → Fuc → Gal-OMe; Fuc(SO_3_ H) → Gal → Fuc-OMe; Gal(SO_3_ H) → Fuc → Fuc-OMe; Fuc → Gal → Fuc(SO_3_ H)-OMe; Fuc → Fuc → Gal(SO_3_H)-OMe; Gal → Fuc → Fuc(SO_3_ H)-OMe.

The fragmentation for the doubly charged ion at *m*/*z* 330.048 (−2) was identified as [Me Gal_2_Fuc (SO_3_H)_2_-2H]^2−^ in Figure 3C. The doubly charged ions at *m*/*z* 314.024 (−2) (indicating a loss of MeOH (32 Da)) and 240.997 (−2) (indicating a loss of methyl fucose (178 Da)) represent a B_3_-type ion and B_2_-type ion, respectively, indicating that the ion at *m*/*z* 330.048 (−2) was Gal(SO_3_H) → Gal(SO_3_H) → Fuc-OMe. The singly charged ion at *m*/*z* 419.076 (−1) was [Me GalFucSO_3_H-H]^−^ derived from the loss of the dehydrated, sulfated galactopyranose residue (226 Da), corresponding to a Y_2_-type ion. The fragment ion at *m*/*z* 403.191 (−1) could be explained through B’_2_ fragmentation, where it would be derived from the loss of the methyl glycoside from sulfated fucose (258 Da). The fragment ions at *m*/*z* 225.003 (−1), 240.997 (−1), 257.010 (−1), and 273.441 (−1) corresponded to [FucSO_3_H-H]^−^, [GalSO_3_H-H]^−^, [Me FucSO_3_H-H]^−^, and [Me GalSO_3_H-H]^−^, respectively, indicating that the sulfate was substituted at the reducing end or non-reducing end. Thus, we speculated that the ion at *m*/*z* 330.048 (–2) was also a mixture of isomers: Gal(SO_3_H) → Gal(SO_3_H) → Fuc-OMe, Gal(SO_3_H) → Gal → Fuc(SO_3_H)-OMe, Gal → Gal(SO_3_H) → Fuc(SO_3_H)-OMe, Gal(SO_3_H) → Fuc (SO_3_H) → Gal-OMe, Gal → Fuc(SO_3_H) → Gal(SO_3_H)-OMe, Gal(SO_3_H) → Fuc → Gal(SO_3_H)-OMe, Fuc(SO_3_H) → Gal(SO_3_H) → Gal-OMe, Fuc → Gal(SO_3_H) → Gal(SO_3_H)-OMe, and Fuc(SO_3_H) → Gal → Gal(SO_3_H)-OMe.

The ESI-CID-MS/MS spectrum of the ion at *m*/*z* 581.136 (−1) ([Me Gal_2_Fuc (SO_3_H)-H]^−^) is shown in Figure 3D. The ions at *m*/*z* 549.129 (−2) (indicating a loss of MeOH (32 Da)), 403.067 (−1) (indicating a loss of methyl fucose (178 Da)), and 241.011 (−1) (a dehydrated sulfated galactose) represent a B_3_-type ion, B_2_-type ion, and B_1_-type ion, respectively; indicating that the ion at *m*/*z* 581.136 (−1) was Gal(SO_3_H) → Gal → Fuc-OMe. The fragment ions at *m*/*z* 435.095 (−1) and 419.099 (−1) could be explained through Y_2_ fragmentation, where they would be derived from the loss of the dehydrated fucose (146 Da) and dehydrated galactose (162 Da), suggesting that both galactose and fucose were at the non-reducing end. The presence of the ion at *m*/*z* 273.044(−1), corresponding to [Me GalSO_3_H-H]^−^, confirmed that the ion at *m*/*z* 581.136 (−1) was a mixture of Gal → Fuc → Gal(SO_3_H)-OMe and Fuc → Gal → Gal(SO_3_H)-OMe. Other isomers could not be excluded. Thus, it was concluded that the ion at *m*/*z* 581.136 (−1) ([Me Gal _2_Fuc (SO_3_H)-H]^−^) was also a mixture of isomers: Gal(SO_3_H) → Gal → Fuc-OMe; Fuc → Gal → Gal(SO_3_H)-OMe and Gal → Fuc → Gal(SO_3_H)-OMe. The fragmentation pattern of the ions at *m*/*z* 873.251 (−1) and 727.193 (−1) was similar to that of the ion at *m*/*z* 581.136 (−1).

The fragmentation pattern of the ions at *m*/*z* 743.188 (−1), 889.244 (−1) and 1035.302 (−1) was also similar to that of the ion at *m*/*z* 581.136 (−1). The ESI-CID-MS/MS spectrum of the ion at *m*/*z* 889.244 (−1) ([Me Gal_3_Fuc_2_ (SO_3_H)-H]^−^) is shown in Figure 3E. The B-type ions indicated that the ion at *m*/*z* 889.244 was a mixture of isomers: Gal(SO_3_H) → Gal → Fuc → Gal → Fuc-OMe and Gal(SO_3_ H) → Gal → Gal → Fuc → Fuc-OMe. The presences of the Y’_4_-type ion and the Y’’_4_-type ion indicated that the non-reducing end was Gal residue and Fuc residue. In addition, the Y1-type ion corresponded to [Me GalSO_3_H-H]^−^. However, other ions could not confirm the sequence of residues. Thus, it was concluded that the ion at *m*/*z* 889.244 was a mixture of 20 isomers (it was emphasized that three continuous Gal residues were present in the backbone of sulfated galactofucan).

From the above discussion, it was concluded that the sulfated galactofucan from *S. thunbergii* contained a backbone of alternating (Gal) _n_ (n ≤ 3) (there were no ions at *m*/*z* 759 ([Me Gal_4_(SO_3_H)-H]^−^) and 905 ([Me FucGal_4_(SO_3_H)-H]^−^)) and (Fuc)_n_ (the number “n” was not determined). 

Autohydrolysis reactions have been used for the structural determination of several polysaccharides [32,33,34,35]. To obtain the precise structure of sulfated galactofucan, an autohydrolysis reaction was performed. Two fractionations (ST-2-S and ST-2-C) were obtained. MS was performed to analyze ST-2-S in Figure 4. It was shown that ST-2-S contained mainly sulfated fuco-oligomers, accompanied by slightly sulfated fuco-xylo-oligomers. The proposed compositions of the ions are also summarized in Table 1. It was proposed that the major components of ST-2-S were the branches of ST-2, while the structure of ST-2-C was the backbone of ST-2 after autohydrolysis. Thus, it was concluded that ST-2 might have branches with sulfated fuco-oligomers.

The ST-2-C was fractionated into ST-2-H and ST-2-L. The chemical analysis of ST-2-H showed that it contained 25.97% Fuc and 26.25% sulfate content. The molecular weight was 158.8 and 32.1 kDa. The molar ratio was 0.12:0.04:0.10:0.07: 0.33:0.13:1.00. (Man:Rha:GlcA:Glc:Gal:Xyl:Fuc). The IR spectrum of ST-2-H (Figure 1C) was similar to that of ST-2, suggesting that ST-2-H had similar functional groups to those of ST-2, (the sulfation of ST-2-C was mainly at the axial C-4 positions). On the other sides, the chemical analysis of ST-2-L showed that it contained 44.50% Fuc and 27.16% sulfate content. The molecular weight was 7.2 kDa, and the molar ratio was 0.17: 1.00 (Gal: Fuc). The IR spectrum of ST-2-L (Figure 1D) was similar to that of ST-2, suggesting that ST-2-L had similar functional groups to those of ST-2 (the sulfation of ST-2-C was mainly at the axial C-4 positions). Thus, it was concluded that both ST-2-H and ST-2-L were sulfated galactofucans. Based on the above speculation, ST-2-C was considered to be the backbone of ST-2. Therefore, Gal residues must be interspersed in the backbone, which was in accordance with the previous study [8]. The molar ratio of sulfate to Hex (Fuc and Gal) residues of ST-2-H was 1.32, suggesting that every Hex (Fuc and Gal) residue has at least one sulfate group. The molar ratio of sulfate to Hex (Fuc and Gal) residues of ST-2-L was 0.89, suggesting that every Hex (Fuc and Gal) residue did not have one sulfate group. However, when the Gal residues did not have sulfate groups (as was confirmed by the NMR results in Figure 5), the molar ratio of sulfate to Fuc residues of ST-2-L was 1.04, suggesting that every Fuc residue has one sulfate group.

The NMR spectra (not shown) of ST-2-H were complicated and unclear because of the high molecular weight. However, the NMR spectra of ST-2-L were recognizable. The spectra (Figure 5) of ST-2-L showed resonances with chemical shifts 99.65 (C-1)/5.11 (H-1), 67.51 (C-2)/3.98 (H-2), 75.95 (C-3)/4.32 (H-3), 80.21–81.04 (C-4)/4.62 (H-4), 68.72 (C-5)/4.32 (H-5), and 15.97/15.71(C-6)/1.09–1.26 (H-6) ppm that were characteristic of 3-linked α-l-fucopyranose sulfated at C4, and weaker resonances with chemical shifts of 61.55 and 65.57 (C-6) that corresponded to 2-linked and 6-linked α-d-galactopyranose, respectively. The proposed structure scheme of ST-2-L was shown in Figure 6. 

Therefore, it was concluded that ST-2 contained various sulfated galacto-fucans. However, they had the same backbone of alternating (Gal)_n_ (n ≤ 3) and (Fuc)_n_. In addition, they also had branches, including sulfated fuco-oligomers, sulfated galacto-oligomers and xylo-oligomers. In a previous study [36], it was found that ST-1 might also contain sulfated galactofucan, which consists of a backbone of alternating (Gal)_n_ and (Fuc)_n_, sulfated randomly on Gal and mainly on C2 in Fuc. Thus, it was hypothesized that *Sargassum thunbergii* might synthesize different kinds of sulfated galactofucan, which consists of a backbone of alternating (Gal)_n_ and (Fuc)_n_ with different molecular weights. In addition, the sulfation pattern was mainly at C4 of fucopyranose and randomly of galactopyranose. Moreover, the branches were mainly sulfated fuco-oligomers.

### 2.2. Anti-Tumor and Anti-Angiogenic Activities of Sulfated Galactofucan and Its Derivates

The anti-tumor and anti-angiogenic activities of ST-2, ST-2-H, and ST-2-L are shown in Figure 7. ST-2-H and ST-2-L were obtained from ST-2. It is shown in Figure 7A that the activities of ST-2-L were similar to those of ST-2, which were higher than ST-2-H. The major differences between ST-2-H and ST-2-L were the molecular weights and the fucose content. In a previous study [22], it was shown that there was no change in the anti-tumor activity at high concentrations (higher than 1 mg/mL). Thus, it was suggested that higher fucose content in sulfated galactofucan leads to better activity. For anti-angiogenic activities, it can be seen in Figure 7B that ST-2-L has the best activities, followed by ST-2 and ST-2-H, suggesting that low molecular-weight sulfated galactofucan, with higher fucose content, has better anti-angiogenic activities.

## 3. Materials and Methods

### 3.1. Materials

The brown algae *Sargassum thunbergii* was collected in Qingdao, China, on 28 May 2014. The l-fucose, d-galactose, d-mannose, d-glucuronic acid, l-rhamnose monohydrate, d-xylose, d-glucose, and 3-methyl-1-phenyl-2-pyrazolin-5-one (PMP) were purchased from Sigma-Aldrich Chemical Co. (St. Louis, Missouri, MO, USA).

### 3.2. Preparation and Purification of Polysaccharides

Polysaccharide (ST) was prepared according to a previous study [36]. Briefly, algae (100 g) were cut into pieces and treated with 85% ethanol three times to remove the pigment. Crude polysaccharide was extracted from the residual material with hot water (3 L) for 4 h. The extract solution was filtered with Celite and concentrated. Further elimination of alginate was achieved using 20% ethanol with MgCl_2_ (0.05 mol/L). After removing the alginate, the supernatant fluid was ultra-filtered. Finally, the dialysate was concentrated, and crude polysaccharide was obtained by ethanol precipitation, namely ST (The yield was 1.30%).

Crude polysaccharide (ST) (6 g) underwent anion exchange chromatography on a DEAE-Bio Gel Agarose FF gel (6 cm × 40 cm, Zhengguang, Hangzhou, China) with elution by 0.5 M (5 L) (ST-1) (the yield was 23.13%) and 2 M (5 L) (ST-2) (the yield was 25.50%). The polysaccharides were then dialyzed, concentrated, and precipitated by ethanol.

### 3.3. Preparation of Desulfated Polysaccharides

The desulfated polysaccharide was prepared according to a previously described method [37]. Briefly, ST-2 (1 g) was dissolved in distilled water (100 mL) and mixed with cationic resin (H^+^) for 3 h. After filtration, the solution was neutralized with pyridinium and lyophilized. The solution was dissolved in dimethyl sulfoxide (DMSO) (St. Louis, Missouri, MO, USA): methanol (9:1; *v*/*v*, 20 mL) and heated at 80 °C for 5 h, and the desulfated product (ST-2-DS) (the yield was 21.69%) was dialyzed and lyophilized. 

### 3.4. Depolymerization of ST-2 by Autohydrolysis

The autohydrolysis was performed according to the modified method [38]. Briefly, ST-2 (0.8 g) was changed to the H^+^ form using a column of cation exchange and left for 72 h at room temperature. The mixture was neutralized with 5% NH_4_OH solution in water. The solution was concentrated and precipitated by ethanol. Finally, two fractionations (the supernatant was named ST-2-S, whereas the precipitate was named ST-2-C) were obtained. ST-2-S was desalted on a Sephadex G-10 column (4.5 × 40 cm, St. Louis, MO, USA), and ST-2-C was fractionated on a Bio-Gel P-10 Gel column (2.6 × 100 cm) eluted with 0.5 M NH_4_HCO_3_ into two fractions, ST-2-H and ST-2-L, and desalted on a Sephadex G-10 column (4.5 × 40 cm, St. Louis, Missouri, MO, USA). The yields of ST-2-S, ST-2-L, and ST-2-H were 11.25%, 16.25%, and 55.03%, respectively.

### 3.5. Compositional Analysis

The sulfated contents were preformed by ion chromatography on a Shodex IC SI-52 4E column (4.0 × 250 mm, Showa Denko K.K., Tokyo, Japan) and eluted with 3.6 mM Na_2_CO_3_ at a flow rate of 0.8 mL/min at 45 °C. The molar ratio of monosaccharides and fucose content was determined as described by Zhang et al. [39]. Briefly, polysaccharides (10 mg/mL) were hydrolyzed by trifluoroacetic acid (2 M) under a nitrogen atmosphere for 4 h at 110 °C. Then, the hydrolyzed mixture was neutralized to pH 7 with sodium hydroxide. Later, the mixture was converted into its PMP derivatives and separated by high performance liquid chromatography (HPLC) chromatography on an YMC Pack ODS AQ column (4.6 × 250 mm, YMC, Kyoto, Japan). Uronic acid (UA) concentration was determined by a modified carbazole method [40]. The molecular weights of the polysaccharides were evaluated by gel permeation chromatography (GPC)-HPLC on a TSK G3000 PWxl column (7 μm 7.8 × 300 mm, TOSOH, Tokyo, Japan), with elution in 0.05 M Na_2_SO_4_ at a flow rate of 0.5 mL/min at 40 °C with refractive index detection. Ten different molecular weight dextrans, purchased from the National Institute for the Control of Pharmaceutical and Biological Products (Beijing, China), were used as weight standards.

### 3.6. IR and MS Analysis of Oligosaccharides

Infrared spectra (IR) were recorded from powder in KBr pellets on a Nicolet-360 fourier transform infrared spectrometer (Nicolet, Pleasanton, CA, USA) between 400 and 4000 cm^−1^ (36 scans, at a resolution of 6 cm^−1^).

ESI-MS and ESI-CID-MS/MS were performed on an LTQ ORBITRAP XL (Thermo Scientific, Waltham, MA, USA) (after installation parameters were modified). The samples were dissolved in CH_3_CN-H_2_O (1:1, *v*/*v*). The solution was centrifuged for 10 min at 10,000 rpm, and the supernatant was analyzed. Mass spectra were registered in the negative ion mode at a flow rate of 5 μL min^−1^. The capillary voltage was set to −3000 V, and the cone voltage was set at −50 V. The source temperature was 80 °C, and the desolvation temperature was 150 °C. The collision energy was optimized between 10 and 50 eV. All spectra were analyzed by Xcalibur.

### 3.7. NMR Spectroscopy

Polysaccharides (50 mg) were co-evaporated with deuterium oxide (99.9%) twice, before dissolving in deuterium oxide (99.9%) containing 0.1 μL deuterated acetone. NMR and two-dimensional spectra were recorded with a Bruker AVANCE III (Bruker BioSpin, Billerica, MA, USA) at 600 MHz and 25 °C. The chemical shifts were adjusted to the internal standard (deuterated acetone, 2.05 and 29.92 ppm, respectively). 

### 3.8. Anti-Tumor and Anti-Angiogenic Activities

Anti-tumor activities of polysaccharides against human lung cancer A549 cells and anti-angiogenic activities against human umbilical vein endothelial (HUVEC) cells were determined. A 3-(4,5-dimethylthiazol-2-yl)-2,5-diphenyl tetrazolium (MTT) assay was used to measure cell viability. Briefly, cells were cultured in RPMI 1640 medium containing 10% fetal bovine serum and penicillin-streptomycin (100 units/mL) in an atmosphere of 5% CO_2_ at 37 °C. The cells (100 µL) were then seeded in a 96-well plate at a density of 1 × 10^4^ cells/well for 24 h. Subsequently, the cells were divided into the following three groups: (1) blank group, which only contained medium for 24 h; (2) control group in which cells were added for 24 h; and (3) experimental groups in which cells and polysaccharides at different concentrations (0.95, 1.82, and 2.61 mg/mL) were cultivated in medium for 24 h and 48 h. After removal of the media, 10 µL of MTT (5 mg/mL) was added to each well. After 4 h of incubation, the supernatants were removed, and dimethyl sulfoxide (DMSO) (100 µL) was added. Next, the absorbance was measured at 490 nm, and the inhibition rate was determined using the following equation: Cell Inhibition rate (%) = (Ac − A1)/(Ac − A0) × 100, where A0 was the absorbance of the blank, A1 was the absorbance in the presence of samples, and Ac was the absorbance of the control.

### 3.9. Statistical Analysis

All data are shown as the mean ± standard deviation (SD). Significant differences between experimental groups were determined by one-way ANOVA, and differences were considered to be statistically significant if *p* < 0.05. All calculations were performed using SPSS 16.0 statistical software.

## 4. Conclusions

In this study, polysaccharide (ST-2) was extracted and fractionated from *S. thunbergii* with hot water. Then, ST-2 was desulfated to obtain ST-2-DS. ST-2-S and ST-2-C were obtained after autohydrolysis. Finally, ST-2-C was fractionated into two fractions (ST-2-H and ST-2-L) by gel chromatography. ST-2-DS and ST-2-S were determined by MS. It was shown that ST-2 contained a backbone of alternating (Gal)_n_ (n ≤ 3) and (Fuc)_n_, branched with sulfated fuco-oligomers. ST-2-L was determined by NMR. It was found that ST-2-L contained 3-linked α-l-fucopyranose sulfated at C4 and interspersed with galactose (the linkages might be 2-linked and 6-linked). Combined with the IR results, it was concluded that ST-2 contained a backbone of (Gal)_n_ (n ≤ 3) and (Fuc)_n_. In addition, the sulfation pattern was mainly at C4 of fucopyranose and randomly at C4 of galactopyranose. Moreover, the branches were mainly sulfated fuco-oligomers. Finally, the anti-tumor and anti-angiogenic activities of ST-2 and its derivates were determined. It was shown that the low molecular-weight sulfated galactofucan, with higher fucose content, had better anti-angiogenic and anti-tumor activities.

## Figures and Tables

**Figure 1 marinedrugs-17-00052-f001:**
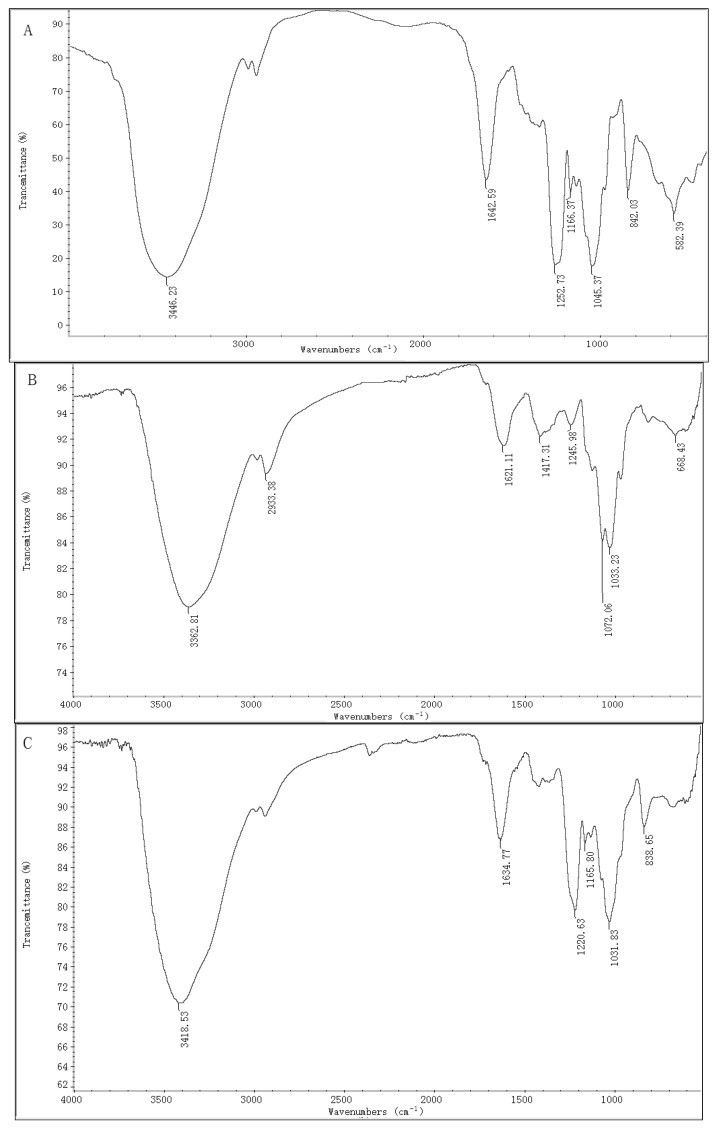
IR spectra of ST-2 (**A**), ST-2-DS (**B**), ST-2-H (**C**), and ST-2-L (**D**).

**Figure 2 marinedrugs-17-00052-f002:**
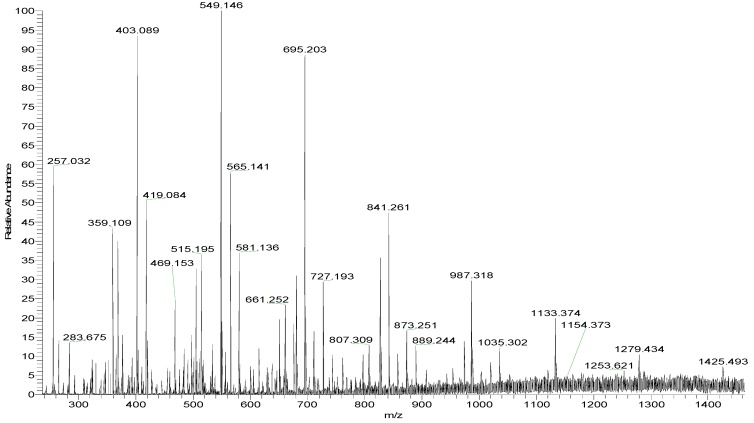
Negative-ion mode ESI-MS spectrum of ST-2-DS.

**Figure 3 marinedrugs-17-00052-f003:**
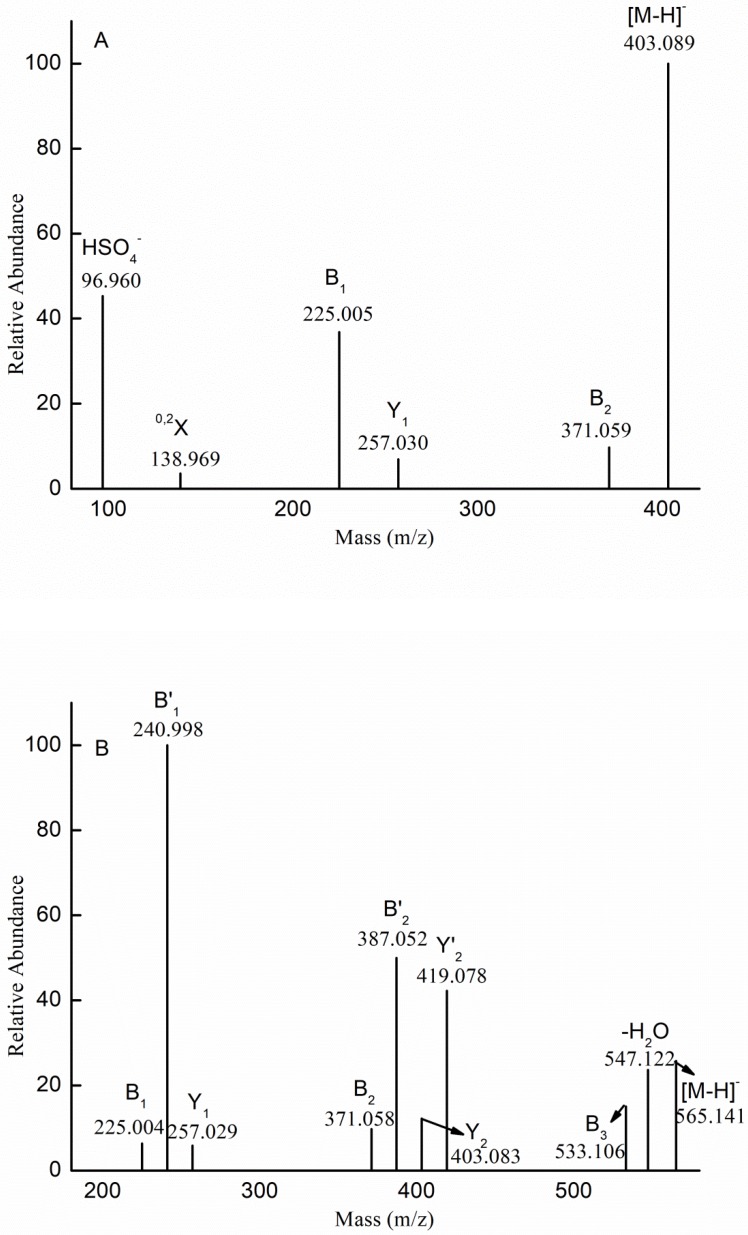
Negative-ion mode electrospray mass spectrometry in tandem with collision-induced dissociation tandem mass spectrometry (ESI-CID-MS/MS) spectra of the ion at *m*/*z* 403.089 (−1) (**A**), 565.141 (−1) (**B**), 330.024 (−2) (**C**), 581.136 (−1) (**D**), and 889.244 (−1) (**E**).

**Figure 4 marinedrugs-17-00052-f004:**
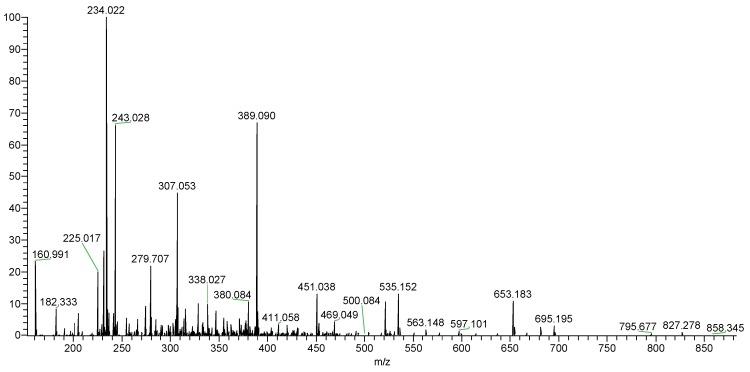
Negative-ion mode ESI-MS spectrum of ST-2-S.

**Figure 5 marinedrugs-17-00052-f005:**
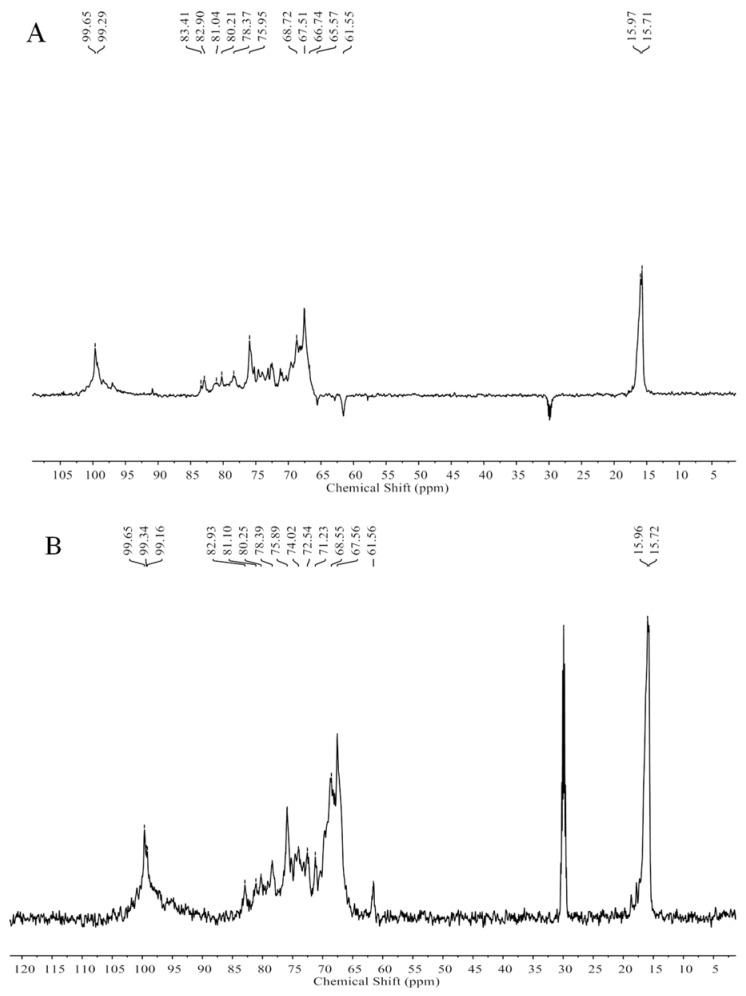
The DEPTQ NMR spectrum (A), the ^13^C NMR spectrum (B), the ^1^H NMR spectrum (C), and the HSQC spectrum (D) of ST-2-L.

**Figure 6 marinedrugs-17-00052-f006:**
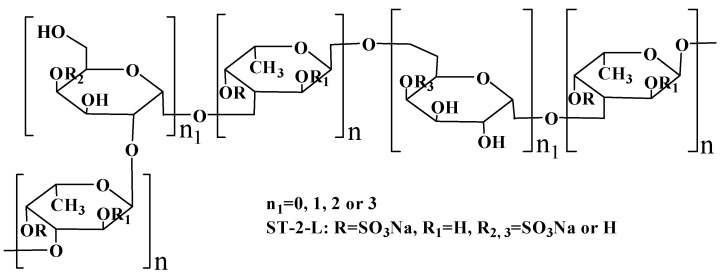
The proposed structure scheme of ST-2-L.

**Figure 7 marinedrugs-17-00052-f007:**
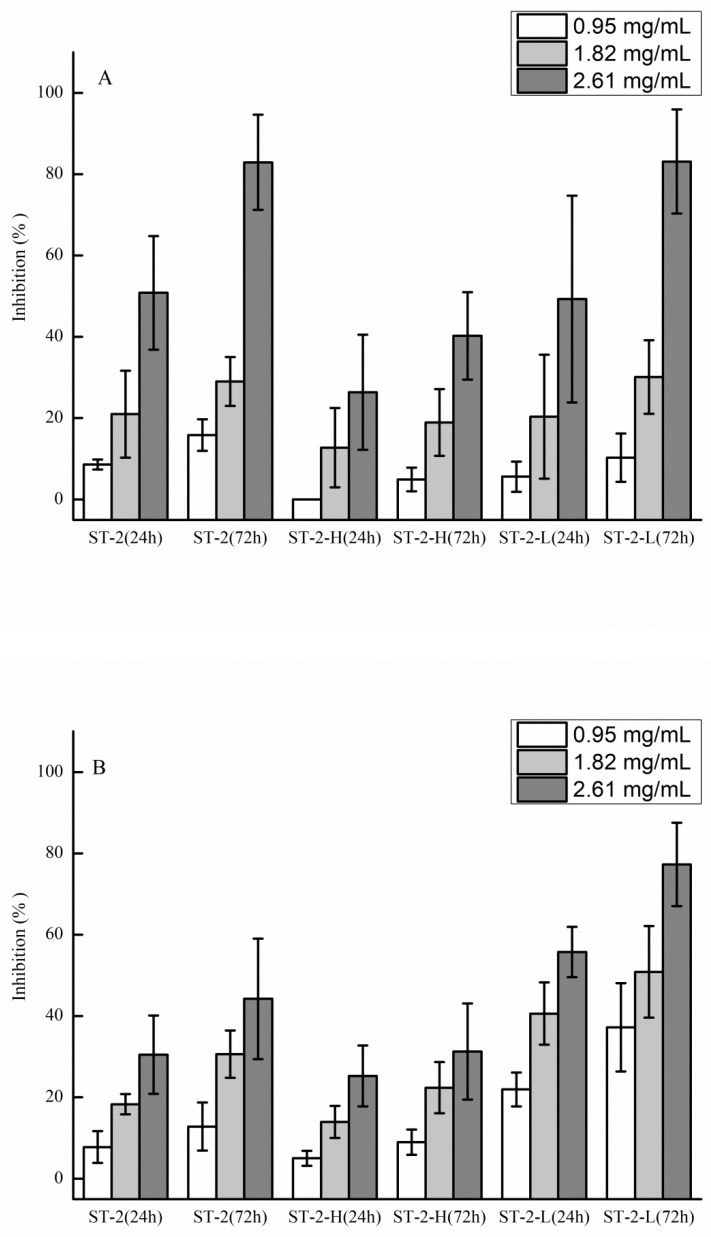
Anti-tumor activities of polysaccharides (ST-2, ST-2-H, and ST-2-L) against human lung cancer A549 cells (**A**) and anti-angiogenic activities against human umbilical vein endothelial cells (HUVEC) (**B**). The results are expressed as percent inhibition. Data are the mean of three determinations +/− SEM.

**Table 1 marinedrugs-17-00052-t001:** Proposed compositions of the ions of desulfated ST-2 (ST-2-DS).

*m*/*z*	Charge	Composition	*m*/*z*	Charge	Composition
257.032	1	[Me Fuc(SO_3_H)-H]^−^	330.024	2	[Me Gal_2_Fuc(SO_3_H)_2_-2H]^2−^
403.089	1	[Me Fuc_2_(SO_3_H)-H]^−^	807.309	1	[Me Gal_2_Fuc_2_(SO_3_H)_2_-H]^−^
549.146	1	[Me Fuc_3_(SO_3_H)-H]^−^	476.059	2	[Me Gal_2_Fuc_3_(SO_3_H)_2_-2H]^2−^
695.203	1	[Me Fuc_4_(SO_3_H)-H]^−^	419.084	1	[Me GalFuc(SO_3_H)-H]^−^
841.261	1	[Me Fuc_5_(SO_3_H)-H]^−^	565.141	1	[Me GalFuc_2_(SO_3_H)-H]^−^
987.318	1	[Me Fuc_6_(SO_3_H)-H]^−^	711.198	1	[Me GalFuc_3_(SO_3_H)-H]^−^
1133.374	1	[Me Fuc_7_(SO_3_H)-H]^−^	857.257	1	[Me GalFuc_4_(SO_3_H)-H]^−^
1279.434	1	[Me Fuc_8_(SO_3_H)-H]^−^	1003.316	1	[Me GalFuc_5_(SO_3_H)-H]^−^
1425.493	1	[Me Fuc_9_(SO_3_H)-H]^−^	581.136	1	[Me Gal_2_Fuc(SO_3_H)-H]^−^
389.073	1	[Fuc_2_(SO_3_H)-H]^−^	727.193	1	[Me Gal_2_Fuc_2_(SO_3_H)-H]^−^
535.130	1	[Fuc_3_(SO_3_H)-H]^−^	873.251	1	[Me Gal_2_Fuc_3_(SO_3_H)-H]^−^
681.188	1	[Fuc_4_(SO_3_H)-H]^−^	1019.305	1	[Me Gal_2_Fuc_4_(SO_3_H)-H]^−^
827.247	1	[Fuc_5_(SO_3_H)-H]^−^	743.188	1	[Me Gal_3_Fuc(SO_3_H)-H]^−^
973.303	1	[Fuc_6_(SO_3_H)-H]^−^	889.244	1	[Me Gal_3_Fuc_2_(SO_3_H)-H]^−^
1119.361	1	[Fuc_7_(SO_3_H)-H]^−^	1035.302	1	[Me Gal_3_Fuc_3_(SO_3_H)-H]^−^
160.991	2	[Fuc(SO_3_H)_2_-2H]^2−^	347.032	2	[Fuc_3_(SO_3_H)_3_-2H]^2−^
182.332	3	[Fuc_2_(SO_3_H)_3_-3H]^3−^	380.084	2	[Fuc_4_(SO_3_H)_2_-2H]^2−^
234.022	2	[Fuc_2_(SO_3_H)_2_-2H]^2−^	469.049	1	[Fuc_2_(SO_3_H)_2_-H]^−^
243.028	1	[Fuc(SO_3_H)-H]^−^	695.195	1	[Fuc_3_(SO_3_H)_3_-H]^−^
279.707	3	[Fuc_4_(SO_3_H)_3_-3H]^3−^	521.136	1	[XylFuc_2_(SO_3_H)-H]^−^
307.053	2	[Fuc_3_(SO_3_H)_2_-2H]^2−^	653.183	1	[Xyl_2_Fuc_2_(SO_3_H)-H]^−^

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
