# Peer review of "Structure Analysis and Anti-Tumor and Anti-Angiogenic Activities of Sulfated Galactofucan Extracted from Sargassum thunbergii"

_marinedrugs, 2019, doi:10.3390/md17010052_

Round 1
Reviewer 1 Report
In the manuscript submitted to Marine Drugs (*) authors works on the structure analysis, anti-tumor and anti-angiogenic activities of sulfated galactofucan extracted from sargassum thunbergii. This reviewer suggest the publication in Marine Drugs after minor revision.
Theme is interesting, techniques are the adequated to solve this kind of analysis, and results are interesting.
Minor comments:
* In Experimental, for Instrumentation, Materials and Reagents, or Programs and Databases (as SPSS, Excel, and others) ever, Product (Manufacturer, City, Country), in this order and format. Please correct in some places. In the case of USA products: Product (Manufacturer, City, State, USA).
* Please describe more acuratly the centrifugation step. Provide the g data, because the same g value can be obtained with differents rpm and times, depending of the centrifuge's radius of different labs.
Author Response
Response to Reviewer 1 Comments
In the manuscript submitted to Marine Drugs (*) authors works on the structure analysis, anti-tumor and anti-angiogenic activities of sulfated galactofucan extracted from sargassum thunbergii. This reviewer suggest the publication in Marine Drugs after minor revision.
Theme is interesting, techniques are the adequated to solve this kind of analysis, and results are interesting.
Minor comments:
Point 1: * In Experimental, for Instrumentation, Materials and Reagents, or Programs and Databases (as SPSS, Excel, and others) ever, Product (Manufacturer, City, Country), in this order and format. Please correct in some places. In the case of USA products: Product (Manufacturer, City, State, USA).
Response 1: Thank you. We have revised according to your comment. Revised texts are marked in red in the paper.
Point 2: * Please describe more acuratly the centrifugation step. Provide the g data, because the same g value can be obtained with differents rpm and times, depending of the centrifuge's radius of different labs.
Response 2: Thank you. The solution was centrifuged for 10 min at 10000 rpm.
Reviewer 2 Report
The authors made polysaccharide (ST-2) which was extracted and fractionated from S. thunbergii with hot water. Then, ST-2 was desulfated to obtain ST-2-DS. ST-2-S and ST-2-C were obtained after autohydrolysis. They have shown that ST-2 contained a backbone of alternating (Gal)n (n≤3) and (Fuc)n branched with sulfated fuco-oligomers and found that ST-2-L contained 3-linked α-L-fucopyranose sulfated at C4 and interspersed with galactose (the linkages might be 2-linked and
6-linked). They also reported the anti-tumor and anti angiogenic activities of ST-2 and its derivates . It was shown that the low molecular-weight sulfated galactofucan with high fucose content is having superior activity.
Author Response
Response to Reviewer 2 Comments
The authors made polysaccharide (ST-2) which was extracted and fractionated from S. thunbergii with hot water. Then, ST-2 was desulfated to obtain ST-2-DS. ST-2-S and ST-2-C were obtained after autohydrolysis. They have shown that ST-2 contained a backbone of alternating (Gal)n (n≤3) and (Fuc)n branched with sulfated fuco-oligomers and found that ST-2-L contained 3-linked α-L-fucopyranose sulfated at C4 and interspersed with galactose (the linkages might be 2-linked and
6-linked). They also reported the anti-tumor and anti angiogenic activities of ST-2 and its derivates . It was shown that the low molecular-weight sulfated galactofucan with high fucose content is having superior activity.
Response 1: Thank you.
Reviewer 3 Report
The manuscript entitled " Structure Analysis, Anti-tumor and anti angiogenic activities of sulfated galactofucan extracted from Sargassum Thunbergii" BY jin W. et al describes the characterization and evaluation of ST-2, ST-2H and ST-2L compounds. Please see comments below:
1) In results section 2.1, please include the structures of the compounds ST-2, ST-2H and ST-2L
for better illustration.
2) Please provide the mass of ST-2-DS using high sensitivity LC/MS or Maldi TOF for complete molecular weight.
3) In table 1, the authors showed proposed composition of the ions. Have you tested the analysis using reference standards to claim the fragmentation process for better clarification. please include them.
4) In fig. 3B, mass is not matching with M-H2O i.e 549.146. please explain.
5) In line 130, Have the authors made any of these isomers and tested their mass spectra, please include them in the manuscript.
6) In line 164, the authors stated that other ions could not confirm the sequence of residues, explain.
7) In line 181, why ST-2-H has two masses i.e 158.8 and 32.1 kDa?
8) since the authors did not differentiate ST-2L and ST-2-H by IR, please provide an evidence by size exclusion chromatography of both or individual runs. Also include the hplc of ST-2L and ST-2-H to differentiate small fragments.
9) In fig 5 (midde bottom), include a proton nmr of sulfated-C4-FUCOPYRANOSE for comparison. Also integrate the region of 5.0 -5.5 ppm of the proton nmr of figure 5.
10) For fig 6, please include the control experiment in both the cell lines. Looks not much potent.
How about anti-tumor and anti-angiogenic activity upto 72 h in both the cell lines. Please include them.
11) In section 3.2, the authors reported very less yield, please repeat the experiment and account for the loss. Did you wash the celite properly?
requested for major revisions.
Round 2
Reviewer 3 Report
The authors response to comments have been reviewed and fond most of them are convincing. However, point 3 (comment 3) response was not convincing though.
Comment 6: the authors have revised the sentence but not very convincing based the results provided.
Please accept the manuscript for publication.